# Are ESG Shares a Safe Haven during COVID-19? Evidence from the Arab Region

**Musaab Mousa** [1] , **Adil Saleem** [2] **and Judit Sági** [3],*

1   Department of Finance, International Business School (IBS), 1031 Budapest, Hungary; mmousa@ibs-b.hu
2   Doctoral School of Economics and Regional Sciences, Hungarian University of Agriculture and Life Sciences, 2100 Godollo, Hungary; saleem.adil@phd.uni-szie.hu
3   Faculty of Finance and Accountancy, Budapest Business School, 1149 Budapest, Hungary
*   Correspondence: sagi.judit@uni-bge.hu

**Abstract:** The world experienced significant changes in its social and economic lives in 2020–21. Major stock markets experienced an immediate decline. This paper attempts to examine the impact of COVID-19 on stock market performance as well as to identify the differences between the responses of ESG stocks and normal stocks to pandemic conditions in the Arab region. Daily time series for three years between March 2019 and March 2021 were collected for the S&P Pan Arab Composite index and S&P/Hawkamah ESG Pan Arab Index. We used a generalized autoregressive conditional heteroscedasticity (GARCH) model to measure market shocks and a non-linear autoregressive distributed lagged (NARDL) regression model to display the relationship between COVID-19 measurements and the performance of stock indexes. The findings suggest that the volatilities of ESG portfolios and conventional ones were equally affected in the pre-COVID period. However, in the post-COVID period, the magnitude of volatility in the ESG stock index was significantly less compared to that of the conventional stock index. The results also revealed that in the ESG market, shock tended to remain for a shorter period. Furthermore, the ESG index was not affected by the number of confirmed cases and deaths. However, evidence of asymmetric long-run cointegration existed between the S&P index and number of cases and deaths. Increases in the numbers of cases and deaths caused a decline in market index, whereas the reverse trends were observed in the retreat of the pandemic.

**Keywords:** ESG; COVID-19 pandemic; capital markets; sustainable finance

## 1. Introduction

The COVID-19 pandemic induced an unprecedented crisis in the global economy. No country was able to escape the serious consequences of the pandemic, and the vast majority of countries experienced partial or full lockdown. Since the World Health Organization announced the virus as a pandemic on 11 March 2020, government measures accelerated to maintain the health system, on the one hand, and to mitigate the economic effects of the crisis on the other. At time of writing, the challenge of vaccine development and deployment is being addressed. By the end of 2020, estimates indicated a contraction in many major economies, with increases in unemployment rates and number of bankruptcies. According to the World Bank, the number of countries experiencing a per capita income decline was the highest since 1870 and there was an economic retraction of 7% in developed countries, compared to 2.5% in developing and emerging economies [1]. Inherently, the COVID-19 pandemic led to a conclusive degree of uncertainty.

As in the real economy and product market, the crisis also affected the financial economy, especially the financial market, which is characterized by high sensitivity to events and relative information, particularly during crisis periods where many events and phenomena such as natural disasters, wars and pandemics have been analyzed to demonstrate a non-financial side of market behavior that contributes to five to thirteen

times greater return volatility [2]. Subsequently, updated news regarding the virus was coupled with considerable shocks in both developed and emerging markets. As happened in other known crises, the current pandemic crisis triggered a panic wave among investors, who opted for heavy selling, especially by the end of the first quarter 2020 [3]. As a result, six trillion USD was lost in the main global market in the last week of February 2020 [4].

Sustainable finance experienced significant growth at the beginning of the twenty-first century in response to the requirements of stakeholders to achieve economic value integrated with considerations of the environment, society and governance (ESG). ESG presents a new measure of value maximization and risk reduction and has become an important part of the evaluation process for companies: the investor relies on it to identify non-financial risks that may pose a threat to the company and its assets. ESG elements have been classified as non-financial factors included in firm value. Thus, the measurement of these elements' impact on firm value is an intricate process that some of the most prestigious specialized companies tackle. Consequently, ESG indexes (e.g., S&P; FTSE) were created to meet the needs of investors who aim to invest in responsible investment channels that are friendly to the environment and society. ESG indexes provide clear examples of the good performance of portfolios formed based on environmental and societal considerations.

The pandemic period highlighted the importance of adopting ESG principles to mitigate the threats of human activities, as well as re-prioritizing the business agenda through a set of initiations and actions. First, many regulators and governments adopted a new approach regarding financial motivations and regulation structures that tilted toward ESG. Second, there was a significant shift in consumption towards clean transport by focusing more on global warming concerns. Third, the global supply chain was restructured to avoid similar risks of supply shortages [5]. Epidemic conditions represent a real experiment in ESG application efficiency, testing the extent to which investors' attitudes to ESG were affected by the repercussions of the pandemic. Therefore, a comparison with normal stocks could show the strengths and weaknesses of ESG investment, especially in a time of crisis. Accordingly, it was assumed that ESG investment would be affected differently than traditional investment by the conditions of COVID-19.

Similarly to other regions around the world, the Arab region suffered from the repercussions of the pandemic, especially with the simultaneous drop in oil prices. The economic outlook for 2020 indicated that GDP declined by 4.2% for oil-exporting countries compared to 0.7% for importing countries in the Middle East and North Africa (MENA) [6]. Likewise, stock markets reacted to pandemic news through the volatility of major indexes, as well as the performance of listed companies. The region's markets are classified as emerging or frontier markets, with some relatively important initiatives in relation to sustainable investment, such as the Dubai financial market ESG index and the Corporate Governance Institute database. The total sustainability-related assets under management were estimated in the MENA region at 54.25 billion USD in 2011 [7]. According to the HSCB survey, 41 percent of investors and corporations in Middle East plan to integrate ESG elements into their operating policies, which refers to a growing awareness of these issues in investment society [8].

Despite the plentiful number of studies into the relationship between financial markets and COVID-19, none of them compared the severity of the impact between traditional shares and ESG shares in the Arab region, which has some ambitious initiatives to promote sustainable finance as an effective investment channel. To bridge this research gap, the current study tried to discover the impact of COVID-19 on stock markets in the Arab region, as well as the differences in terms of impact on traditional and ESG portfolios using the methodology of the GARCH model to measure market shock caused by the pandemic. It also employed the ARDL model to analyze the impact of COVID-19, measured by affected cases and fatalities, on both the normal index and ESG index to determine differences in relation to COVID-19 impact.

This paper covers the related literature concerning the COVID-19–capital market relationship in normal and ESG portfolios in the second part. The third part includes

a description of the applied methodology and data collection, while the fourth part covers the statistical results and a discussion of our major findings.

## 2. Literature Review

### 2.1. Stock Markets and COVID-19

Intrinsically, stock markets translate phenomenal and substantial events into performance movement, such as firm value or price volatility. Many empirical studies have proved the interconnection between stock markets, political events [9], natural disasters [10], and wars [11], as well as earlier epidemics such as Ebola [12] and SARS [13,14]. Studies agree that investor panic leads to anomalous market reactions, which also applied to the COVID-19 repercussions, as the Chicago Board Options Exchange's CBOE volatility index recorded a historic figure of 80 points during the early spread of the pandemic, which exceeded the peak during the 2008 financial crisis. The market reaction was much greater compared to other epidemics that have spread in the past [15] due to a decline in profitability due to lockdowns, as well as the rapid spread of financial contagion and economic shocks due to high global financial integration of markets [16]. Accordingly, a considerable body of research has attempted to investigate the impact of COVID-19 on stock market performance, whether in specific markets or in international or regional markets. Table 1 displays some related literature regarding the bilateral relationship between the stock exchange and COVID-19 with regard to market scope, variables and methodology. Reference [17] investigated stock market reactions to the number of confirmed cases and fatalities in 64 countries, and the results indicated that the markets responded more negatively to the number of confirmed cases than the number of fatalities in terms of return. In the same manner, ref. [18] examined the impact of COVID-19 on 53 emerging markets and 23 developed markets, and the results concluded that pandemic cases and fatalities led to lower returns, higher volatility, and lower trading volume in emerging markets during the growing contagion period. In contrast, only cases played this role in developed markets during the period of stabilizing spread, which suggests differences in investor reactions to the pandemic event in the two groups of markets. From a market connectedness perspective, the COVID-19 shock hit the main regional markets, as seen in the daily return volatilities of MSCI indexes, due to a high level of interdependence (except for the Latin American index) [19]. In addition, investors' fears played a pivotal role in transferring the negative reaction to confirmed cases in the main markets of WHO regions, with higher negative and abnormal returns in Western Pacific markets compared to other territories [20]. Focusing on regional markets, ref. [21] analyzed the effect of the pandemic measured via two confirmed variables, number of cases and time since first reported case, on capital market indexes in Central, North, and South America. The results confirmed the adverse impact of first variable, while the time since the first reported case positively affected the studied indexes. By analyzing 29 emerging markets based on MSCI classification, ref. [22] showed that the impact of COVID-19 became less negative after mid-April 2020 and that the impact was higher in Asian markets compared to European emerging markets. In relation to nine Islamic stock indexes, ref. [23] concluded that most of these indexes had been affected by WHO's declaration of a global health emergency with a long-term trend of volatility shocks. In China, where the virus was first discovered, all listed companies' returns correlated negatively with growing confirmed cases and fatalities [24]. Furthermore, share returns varied during the pandemic depending on the beneficiary and affected sectors. Whereas food and healthcare stocks achieved exceptionally positive returns, oil, real estate, entertainment and hospitality stocks suffered negative returns [25]. Correspondingly, in the MENA region, the effect of COVID-19 was measured by confirmed cases, deaths, and stringency index, which had a significant impact on liquidity on both the market and company level, regardless of size and sector [26]. Likewise, the markets' reaction in the region varied between being influenced by the cumulative number of deaths or the cumulative number of cases [27]. Based on what has been covered in the relevant literature, the consequences of COVID-19 on the markets were not limited to a specific market or region, but rather were a global

phenomenon as the pandemic spread. Consequently, based on the related literature, the first hypothesis was as follows:

**Table 1.** Summary of some related literature.

| Study | Market Scope | Relationship | Methodology |
|---|---|---|---|
| Ashraf (2020) | 64 emerging and developed countries | [Number of confirmed cases and fatalities] and [market return] | Panel regression |
| Harjoto et al. (2020) | 76 emerging and developed countries | [Number of confirmed cases and fatalities] and [market return, volatility, and volume] | Multivariate regressions |
| Al-Qudah and Houcine. (2021) | 6 major affected WHO Regions (Africa, Americas, Eastern Mediterranean, Europe, South-East Asia and Western Pacific) | [Number of confirmed cases] and [daily stock return] | Panel regression |
| Amin et al. (2020) | Central, North, and South America. | [Number of confirmed cases and age] and [market index] | Panel regression |
| Topcu and Gulal (2020) | 26 emerging markets | [Number of confirmed cases, exchange rates, and oil price shocks] and [market return] | Panel regression |
| Al-Awadhi et al. (2020) | Chinese listed companies | [Daily growth of cases and fatalities] and [return of stock] | Panel regression |
| Saleem et al. (2021) | Nine Islamic indexes | Volatility of index reaction to COVID-19 | Event study and GARCH (1,1) |
| Mdaghri et al. (2021) | Listed companies on six markets of MENA | [Daily growth of cases and fatalities] and [liquidity and effective spread of share] | Panel regression |
| Current study | Arab region normal index and ESG index | [Number of confirmed cases and fatalities and daily new number of cases and fatalities] and [indexes return] | GARCH model & Non-linear ARDL |

**Hypothesis 1 (H1).** *The COVID-19 pandemic significantly affects stock performance in the Arab region.*

*2.2. ESG and COVID-19*

ESG investment began to grow significantly when the global investment market noticed the increasing need for products geared towards what became known as the responsible investor. It aims to sustain returns associated with positive, long-term outcomes for the environment, society, and business.

ESG expenditure and related disclosure play a significant role in value creation in the market, which is in line with the value enhancing theory. Several scholars have provided scientific proof of the positive impact of ESG and related disclosure on shareholders' value using common valuation models [28,29]. In a global framework, ref. [30] concluded that ESG contents presented a reliable predictor of share performance based on constituents of the MSCI All Country World Investable Market Index (ACWI IMI). In relation to COVID-19, high ESG portfolio performance exceeded low ESG portfolio performance, and ESG stan-

dards reduced financial risk during the COVID-19 crisis [31]. By examining a worldwide sample of more than 6000 stocks in 45 countries during 1Q 2020, Gianfrate et al. [32] concluded that corporate stocks with higher ESG ratings had higher returns only when their home countries' stocks in general had higher abnormal stock returns. Thus, their argument gave way to regional analysis, finding that ESG stocks did improve over conventional stocks in North America, and recommending further empirical studies concerning ESG stocks' heterogeneous responses to the pandemic. The same applied to the ESG Exchange-Traded Fund (ETF), which maintained higher returns than the market [33], meaning that investment in ESG could be an effective method of avoiding the high volatility in prices during the crisis [34]. This was confirmed from a risk hedging perspective, where investment decision-making based on ESG risk was perceived to be an effective strategy in the early phase of the COVID-19 crisis [35]. Furthermore, ESG factors increased the explanatory power of the three factors of the Fama–French model during the first quarter of 2020 [36].

In summary, ESG is a promising approach for investment valuation, especially after the COVID-19 crisis, as this concept was put under the microscope and the pace of experimental studies seeking to explain its role in overcoming crisis risk compared to other investment channels has increased. However, our study does not include the presumption that socially responsible investors drive companies to implement measures and this is why they become more resilient to external shocks [37,38], nor that such behavior allows ESG-oriented investors to be more resilient in holding their stocks and avoid fire sales [39]. Accordingly, the second hypothesis can be stated as the following:

**Hypothesis 2 (H2).** *The ESG index is significantly affected by the consequences of COVID-19, and differently from the normal index in the Arab region.*

## 3. Materials and Methods

The aim of this study is to explore the impact of the COVID-19 crisis on capital market performance in the Arab region on one hand and to compare normal stock performance with ESG performance on the other. To address this goal, data on capital markets were collected using the S&P database for three years between March 2019 and March 2021. The daily values of the S&P Pan Arab Composite index (S&P Normal) were used to measure capital market performance and daily values of the S&P/Hawkamah ESG Pan Arab Index (S&P ESG) to measure ESG performance. The latter index includes the 50 highest scoring companies in terms of ESG criteria in the 11 major Arab markets. COVID-19 data, including daily confirmed cases and death numbers, were obtained from the Worldometer database provided by the World Health Organization (WHO, Geneva, Switzerland).

Our methodology consists of two parts. The first part explains the volatility changes in selected market indexes before and after the COVID 19 crisis, while the second part captures the effects of the growth in the number of cases and deaths on the performance of the indexes under study.

### 3.1. Market Shock Measurement—GARCH Model

With regard to the first part, the generalized autoregressive conditional heteroskedasticity (GARCH) model was employed to analyze market shocks caused by the COVID-19 outbreak. GARCH was improved by [40] to overcome the disadvantages of the ARCH (p) model when the model is over-parameterized, which gives inconsistent results and produces negative coefficients. According to [40], in the GARCH model the variance in the financial time series is not only in past squared residuals but also the lagged variance itself. Consequently, many econometricians recommend GARCH (p,q) models for analyzing the volatility of financial time series effectively [41]. Moreover, GARCH (1,1) models provide robust and efficient evidence of volatility changes and their persistence.

To model the volatility effect and persistence caused by COVID-19, we divided the data sets into three different groups. The first group consisted of the whole period under study from 3 September 2019 to 23 March 2021. Based on the date on which COVID-19

was declared a pandemic, this data series was then divided into two additional groups, i.e., pre-COVID group (3 September 2019 to 10 March 2020) and post-COVID group (11 March 2020 to 23 March 2021).

Daily market return was calculated using Equation (1), whereas the mean equation was modeled through the ordinary least square (OLS) method, represented in Equation (2). Equation (3) represents the precondition of ARCH effects, which illustrated that the errors were conditionally distributed with variance $h_t$. Equation (4) describes the GARCH (1,1) process in which variance of the selected series depends on lagged squared residuals and their variance to model the volatility effects and persistence caused by COVID-19.

$$y_{i.t} = \ln\left(\frac{MI_{i.t}}{MI_{i.\ (t-1)}}\right) \tag{1}$$

$$y_{i.t} = \mu_t + \theta_i y_{i.t-1} + \epsilon_t \tag{2}$$

$$\frac{\epsilon_t}{y_{t-1}} \sim N(0,\ h_t) \tag{3}$$

$$h_t = \alpha_i + \sum_{i=1}^{p} \beta_i (\epsilon_{t-i})^2 + \sum_{i=1}^{q} \gamma_i h_{t-i} \tag{4}$$

where $y_{i.t}$ is the market return of $i$ index at time $t$; $MI_{i.t}$ and $MI_{i.\ (t-1)}$ are stock index value of $i$ index at time $t$; $\epsilon_t$ represents the errors generated through Equation (2), which are dependent on conditional variance. The GARCH (p,q) model has lags of $p = 1$ and $q = 1$, as illustrated in Equation (4). The conditional variance of the financial time series is dependent on the square of its lagged residuals $\epsilon_{t-i}$ and its own lag $h_{t-i}$.

In order to ensure positive variance of the series, all three parameters of the GARCH (1,1) model must satisfy the given condition, i.e., $\alpha_i > 0$, $\beta_i \geq 0$, and $\gamma_i \geq 0$. However, the sum of $\beta_i + \gamma_i$ shows the persistence measurement of current shocks in the market. If the sum of the ARCH and GARCH coefficients is not significantly different from 1, the shock will likely remain for a longer period in the financial market [37]. Conversely, if the value of $\beta_i + \gamma_i$ is significantly less than 1, there will be a shorter impact of external shocks. Furthermore, according to [42], the measure of volatility persistence is also referred to as the response function of the market to bear external shocks. On the other hand, results become explosive if the response function exceeds the level of one [43].

### 3.2. COVID-19 Impact Measurement—ARDL Model

The extended version of the autoregressive distributive lagged (ARDL) regression developed by [44] was applied to analyze the performance of both markets in response to the number of new cases and deaths per day. To capture the asymmetric non-linear relationship between regressors, an extended version of ARDL was used, as developed by [45], known as non-linear autoregressive distributive lagged (NARDL). Compared to linear ARDL, asymmetric ARDL shows several advantages that make it more robust and reliable in generating consistent results. For instance, NARDL exhibits appropriate properties even with a small sample. Furthermore, according to [46], the NARDL model is free from several statistical problems, including omitted lag bias and residual correlation. The model enables the capture of long- and short-run asymmetries simultaneously, with the help of partial positive and negative sums of the time series [45] The partial positive and negative sum of the regressors is calculated through the following equations.

$$NC^+{}_t = \sum_{j=1}^{t} \Delta NC^+{}_j = \sum_{j=1}^{t} \max(\Delta NC^+{}_j,\ 0) \tag{5}$$

$$NC^-{}_t = \sum_{j=1}^{t} \Delta NC^-{}_j = \sum_{j=1}^{t} \min(\Delta NC^-{}_j,\ 0) \tag{6}$$

$$ND^+{}_t = \sum_{j=1}^{t} \Delta ND^+{}_j = \sum_{j=1}^{t} \max(\Delta ND^+{}_j,\, 0) \tag{7}$$

$$ND^-{}_t = \sum_{j=1}^{t} \Delta ND^-{}_j = \sum_{j=1}^{t} \min(\Delta ND^-{}_j,\, 0) \tag{8}$$

$$M(var)^+{}_t = \sum_{j=1}^{t} \Delta M(var)^+{}_j = \sum_{j=1}^{t} \max(\Delta M(var)^+{}_j,\, 0) \tag{9}$$

$$M(var)^-{}_t = \sum_{j=1}^{t} \Delta M(var)^-{}_j = \sum_{j=1}^{t} \min(\Delta M(var)^-{}_j,\, 0) \tag{10}$$

$NC^+{}_t$ represents the positive change in the number of confirmed cases on day $t$; $NC^-{}_t$ shows the negative partial sum of decreasing number of cases at time $t$. $ND^+{}_t$ is the increase in new deaths, and $ND^-{}_t$ represents a decrease in new deaths. $M(var)^+{}_t$ is the positive change in market volatility measured through GARCH (1,1) in the previous section. Once the partial positive and negative sums are calculated, we modeled the non-linear ARDL, following [45], using the given equations.

$$\Delta(MI)_{i.t} = \theta_0 + \sum_{q=1}^{p1} \vartheta_{1q}\Delta(MI)_{t-q} + \sum_{q=0}^{p2} \vartheta_{2q}{}^+\Delta NC^+{}_{t-q} + \sum_{q=0}^{p3} \vartheta_{2q}{}^-\Delta NC^-{}_{t-q} + \sum_{q=0}^{p3} \vartheta_{3q}{}^+\Delta M(var)^+{}_{t-q}$$
$$+ \sum_{q=0}^{p3} \vartheta_{3q}{}^-\Delta M(var)^-{}_{t-q} + \omega_1 MI_{t-1} + \omega_2 NC^+{}_{t-1} + \omega_3 NC^-{}_{t-1} + \omega_4 M(var)^+{}_{t-1}$$
$$+ \omega_5 M(var)^-{}_{t-1} + \varepsilon_t \tag{11}$$

$$\Delta(MI)_{i.t} = \theta_0 + \sum_{q=1}^{p1} \vartheta_{1q}\Delta(MI)_{t-q} + \sum_{q=0}^{p2} \vartheta_{2q}{}^+\Delta ND^+{}_{t-q} + \sum_{q=0}^{p3} \vartheta_{2q}{}^-\Delta ND^-{}_{t-q} + \sum_{q=0}^{p3} \vartheta_{3q}{}^+\Delta M(var)^+{}_{t-q}$$
$$+ \sum_{q=0}^{p3} \vartheta_{3q}{}^-\Delta M(var)^-{}_{t-q} + \omega_1 MI_{t-1} + \omega_2 ND^+{}_{t-1} + \omega_3 ND^-{}_{t-1} + \omega_4 M(var)^+{}_{t-1}$$
$$+ \omega_5 M(var)^-{}_{t-1} + \varepsilon_t \tag{12}$$

where $MI_{i.t}$ is the market return of $i$ index at time $t$; $p_i$ represents the optimal numbers of lags for dependent and independent variables. The error term is denoted as $\varepsilon_t$. The first difference of the variables is represented as $\Delta$. $NC^+{}_t$ represents the positive changes in the number of confirmed cases on day $t$; $NC^-{}_t$ shows the negative partial sum of decreasing number of cases at time $t$. $M(var)^+{}_t$ represents the positive change in market volatility. The null hypothesis of the NARDL bound test is tested as $H_0 : \omega_1 = \omega_2 = \omega_3 = \omega_4 = \omega_5 = 0$, whereas rejection of the null hypothesis would be desirable for the presence of long-run asymmetric cointegration among the variables of interest. The values computed for F-statistics in the bound tests are compared with the asymptotic values of F-statistics of the upper and lower bound values [43,47]. Depending on the asymptotic value, if the computed F-statistics are greater than the upper bound I(1) critical value, the presence of asymmetric cointegration is confirmed. However, if the value falls below the lower bound critical value, we cannot reject the null hypothesis of no cointegration. Furthermore, inconclusive results may occur if the F-statistics fall between the lower and upper bound critical values. The long run coefficients of positive and negative regressors were calculated as $-\frac{\omega_2}{\omega_1}$ and $-\frac{\omega_3}{\omega_1}, -\frac{\omega_4}{\omega_1}, -\frac{\omega_5}{\omega_1}$ for $NC^+, NC^-, ND^+, ND^-,\ M(var)^+{}_t M(var)^-{}_t$ respectively. Furthermore, in order to ascertain the asymmetry in long run coefficients of positive and negative partial sums, the Wald test was applied with the null hypothesis $H_o = -\frac{\omega_2}{\omega_1} = -\frac{\omega_3}{\omega_1}$ and $H_o = -\frac{\omega_4}{\omega_1} = -\frac{\omega_5}{\omega_1}$. Similarly, asymmetry in the short run coefficient was tested using Wald tests as $\sum_{i=0}^{q} \vartheta_{2q}{}^+ = \sum_{i=0}^{q} \vartheta_{2q}{}^-$ and $\sum_{i=0}^{q} \vartheta_{3q}{}^+ = \sum_{i=0}^{q} \vartheta_{3q}{}^-$. Furthermore, the robustness of the model was evidenced through various diagnostic tests. Serial correlation was diagnosed through the LM Breusch–Godfrey test. The LM Breusch–Pegan test was performed to

ensure the homoscedasticity of errors. Importantly, the correctness of functional form of the model was diagnosed through the Ramsey RESET test. For the optimal lag selected in ARDL regression, Akaike Information Criteria (AIC) was used with lag *i*. AIC ran $(m+1)^4$ lagged models, where *m* denotes the max number of lags in the regression.

## 4. Empirical Results and Discussion

This section provides the results and discussion based on the econometric models used in this study, first by analyzing volatility to explore market shocks caused by the COVID-19 pandemic and second through ARDL regression analysis to measure the impact of COVID-19 variables on the returns of the indexes under study.

### 4.1. Volatility Results

Table 2 presents the descriptive statistics of the whole, pre-COVID, and post-COVID periods. The mean returns of the SEMGPCPD and SPHMPAUP indexes were positive, which indicates that investors of both indexes earned positive return on average. The deviations from mean return in both markets were almost the same. However, the mean and median values in the SPHMPAUP were lower than those in the SEMGPCPD. This indicates that during the whole period of study from 3 September 2019 to 23 March 2021, the traditional S&P pan-Arab index produced a better return on average compared to the ESG index. Furthermore, in the pre-COVID period, i.e., 3 September 2019 to 10 March 2020, the return on both markets appeared negative with standard deviations of 0.0083 and 0.0067 in the normal and ESG indexes respectively. This indicates that due to fluctuations in the daily return, the two indexes bore negative returns on average. However, the mean return became positive in the post-COVID period in both indexes.

**Table 2.** Descriptive statistics of index returns.

| Period | Index | Mean | Median | St. Dev. | Kurtosis | Jarque Bera | ADF |
|---|---|---|---|---|---|---|---|
| Whole period | S&P Normal | 0.00014 | 0.00015 | 0.00812 | 32.796 | 30,755.71 | 0.000 |
| | S&P ESG | 0.00005 | 0.00010 | 0.00780 | 58.025 | 101,002.1 | 0.000 |
| Pre-COVID | S&P Normal | −0.00032 | 0.00000 | 0.00832 | 41.763 | 30,814.15 | 0.000 |
| | S&P ESG | −0.00035 | −0.00010 | 0.00671 | 59.410 | 64,881.87 | 0.000 |
| Post-COVID | S&P Normal | 0.00082 | 0.00044 | 0.00777 | 14.976 | 2119.847 | 0.000 |
| | S&P ESG | 0.00063 | 0.00035 | 0.00922 | 49.527 | 29,339.44 | 0.000 |

The time series in all three periods were leptokurtic, as the value of kurtosis was more than its normal value (i.e., 3.0). In addition, the stationarity of the data for the three periods was checked by using the augmented Dickey–Fuller (ADF) test, which showed that the data were stationary at level. Normality of all three time periods was not evident, as Jarque–Bera statistics were significant at 1%. Following [43], to model the time series for volatility using ARCH/GARCH, the pre-conditions of leptokurtic, non-normality, and stationarity in the time series was evident from the given statistics. Furthermore, we found evidence of volatility clustering from all three periods as shown in Figure 1.

In order to apply the GARCH (1,1) model, the presence of the ARCH effect should be evident as a precondition, as in Table 3, which shows that ARCH effects were present in the time series for the three periods, as the probability according to the ARCH test was significant at 1% in most periods and at 5% in the post-COVID period for the S&P Pan Arab Composite index.

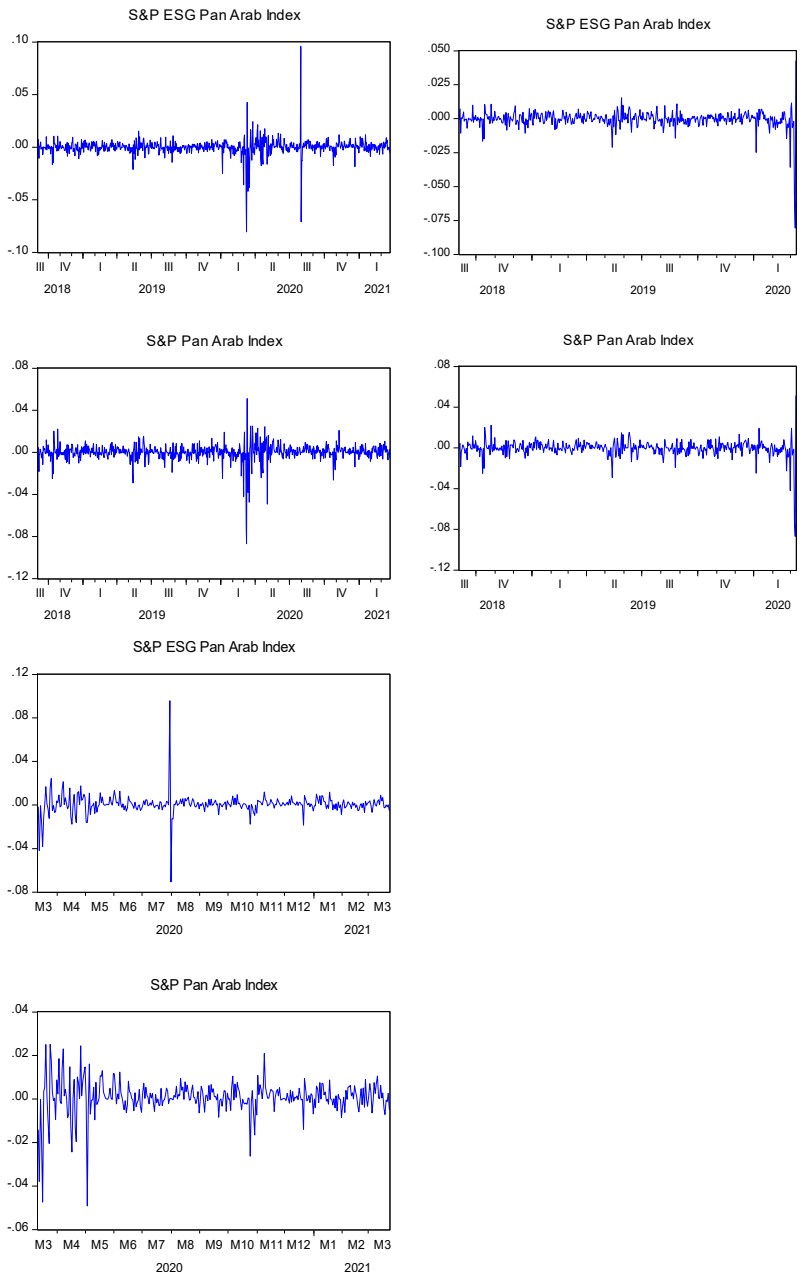

**Figure 1.** Volatility of the S&P Pan Arab index and S&P ESG Arab index during the three periods.

**Table 3.** ARCH Effect of the three periods under study.

| Period | Index | Mean | Median | St. Dev. |
|---|---|---|---|---|
| Whole period | S&P Normal | 395.82 | 0.000 | Present |
| | S&P ESG | 181.25 | 0.000 | Present |
| Pre-COVID | S&P Normal | 271.83 | 0.000 | Present |
| | S&P ESG | 260.07 | 0.000 | Present |
| Post-COVID | S&P Normal | 4.642 | 0.032 | Present |
| | S&P ESG | 42.149 | 0.000 | Present |

The estimates of the GARCH (1,1) model are presented in Table 4, to validate the robustness of the models used in GARCH (1,1). We performed the ARCH LM test to examine the remaining ARCH effects in the time series. Additionally, for autocorrelation we performed a correlogram of residuals tests with 24 lags.

**Table 4.** GARCH (1,1) results.

| Period | Index | $\alpha$ | $\beta$ | $\gamma$ | $\beta + \gamma$ | ARCH-LM | Q24 |
|---|---|---|---|---|---|---|---|
| **Whole period** | S&P Normal | 0.000003 (2.7896) [a] | 0.1404 (2.484) [b] | 0.7700 (11.288) [a] | 0.910 | 0.0145 (0.904) | 13.087 (0.965) |
| | S&P ESG | 0.000002 (5.2828) [a] | 0.1176 (7.9810) [a] | 0.8417 (42.946) [a] | 0.959 | 0.0745 (0.785) | 19.404 (0.496) |
| **Pre-COVID** | S&P Normal | 0.000002 (3.3247) [a] | 0.1443 (5.7784) [a] | 0.8130 (19.846) [a] | 0.957 | 0.2968 (0.585) | 16.960 (0.766) |
| | S&P ESG | 0.000003 (3.2267) [a] | 0.1695 (5.4544) [a] | 0.8082 (21.713) [a] | 0.978 | 0.0365 (0.848) | 23.078 (0.340) |
| **Post-COVID** | S&P Normal | 0.000010 (2.3648) [b] | 0.2599 (1.4800) | 0.5589 (3.590) [a] | 0.808 | 0.0002 (0.989) | 6.3835 (1.00) |
| | S&P ESG | 0.000001 (11.7047) | 0.0009 (0.2257) | 0.9568 (201.45) [a] | 0.957 | 0.2889 (0.590) | 20.546 (0.424) |

Notes: [a,b] denotes significance at 1% and 5% respectively. Q(24) denotes Box–Ljung statistics for 24-order serial correlation of the residuals.

Our results show that the volatility of both indexes was significant in the whole period, and the volatility clustering was evident from the GARCH coefficient in both indexes. The GARCH (1,1) model for both indexes was modelled based on Schwarz information criteria (SIC), adjusted R squares, and maximum likelihood values. Furthermore, for the ESG index in the entire and post-COVID period, all the financial time series were modelled on GARCH (1,1) using a normal gaussian error distribution. The findings suggested that the magnitude of volatility clustering of returns was higher in the traditional S&P Pan Arab index compared to the S&P ESG index. Importantly, the coefficient of ARCH was significantly less than one, which implies that the risk associated with both indexes was not explosive over the entire period. However, in the pre-COVID period the volatility of both indexes remained the same. The volatility of the ESG index was slightly lower than that of the conventional S&P index, but both were statistically significant at 1%. In general, in the pre-COVID period both indexes' volatility/risk behaved in a similar manner, which is confirmed by the magnitudes of volatility clustering of 0.812 and 0.808 respectively. The absence of a significant difference between the two indexes in terms of volatility before the pandemic indicated that investors in the Arab region did not pay attention to ESG shares and dealt with them similarly to other shares in a normal time period. In addition, both sectors performed in a similar fashion in terms of average return and risk associated. However, our findings indicate that the risk profiles of both portfolios reacted differently after the WHO announced COVID-19 as a global pandemic.

The statistical outputs refer to a significant change in volatility clustering of both indexes in the post-COVID period, and the ESG index performed significantly better as compared to the normal S&P Arab index. The magnitude of the GARCH coefficient significantly differed from one in the case of the ESG index, while volatility clustering did not differ from one for the S&P Arab index.

Our results imply that the S&P Pan Arab index was significantly affected in the post-COVID period. The volatility of the S&P during this period significantly differed compared to its volatility during the pre-COVID period, as the GARCH coefficient increased from 0.808 to 0.957. The magnitude of the GARCH coefficient suggests that after COVID-19 was declared a global pandemic, the market of the conventional S&P Arab index became riskier, as the results suggest a huge variation in the error term. However, the same was not true for the ESG index. Our results provided evidence of stability in the ESG Arab index in the post-COVID period. The GARCH coefficient decreased from 0.813 to 0.559, and both values were significant at 1%. This implies that the ESG Arab index proved to be safer in the post-COVID period. Moreover, the model is stable and robust in both pre- and post-COVID periods, as the ARCH-LM and Box–Ljung statistics for serial correlation of the residuals up to 24 lags were insignificant. Therefore, after the markets were hit by COVID-19, the ESG index appeared to be safer for investors. Our finding supports the evidence from [30], which indicated that during the crisis ESG stocks in China performed better and were considered immune to this global crisis.

For deeper analysis, the response function to external shocks was measured by the sum of the ARCH and GARCH coefficients to explain volatility persistence—in other words, the extent to which current shock will prevail in the market. The closer the value of $\beta + \gamma$ to one,

the longer the shock would remain in the market. Across the full period, volatility shocks tended to remain for a longer period in the S&P Arab Index compared to the ESG index. However, the overall response function of both indexes was increased in the pre-COVID period. For instance, in the case of the S&P Arab Composite index, the volatility shock declined from 0.959 to 0.286 ($0.9592^{30} = 0.286$) in the first 30 days after the pandemic was declared. In the first 45 days of the pandemic, the S&P Arab index remained highly volatile with a persistence measurement of 0.1534 ($0.9592^{45}$). The S&P ESG index performed better with a persistence value of 0.9104 to 0.0598 in the first 30 days. The value of the volatility shock significantly declined after 45 days. Overall, the S&P ESG Index showed less risk and lower persistence values for external shocks, and in the post-COVID period, the persistence function of the ESG index declined significantly. The magnitude of volatility shock after the first 30 days decreased significantly ($0.808^{30} = 0.0016$). Comparing the first months of the pre- and post-COVID periods, the shock persistence in the ESG index substantially declined from 0.286 to 0.0016. Importantly, the S&P index produced higher persistence measurements, i.e., 0.977 in the pre- and 0.957 in the post-COVID period. Our findings related to volatility and shock persistence suggest that the ESG index appeared to be safer and less risky compared to conventional S&P Arab Index.

The results regarding volatility and market shock after the pandemic announcement are in line with related literature regarding the lower risk level of ESG shares compared to other shares during the time of the pandemic [32]. ESG may also be considered a safe haven for ensuring stable investment in practical terms when any crisis with a high degree of uncertainty begins [33].

### 4.2. Performance of Market in Connection to COVID-19

The long-run and short-run asymmetric impact of the COVID-19 crisis on selected stock indexes was examined by using a non-linear ARDL bounds testing approach between the logged market index as a dependent variable and the logged numbers of new cases and deaths as an independent variable. However, variance forecasted using the GARCH (1,1) model was also used as an independent variable to predict index performance. We modeled both indexes with the number of new cases and number of new deaths separately from respective market risk, i.e., market volatility. This non-linear model is specified by the following equations.

$$\text{MI}_i = F\left(NC^+, NC^-, M(var)^+_t M(var)^-_t\right) \tag{13}$$

$$\text{MI}_i = F\left(ND^+, ND^-, M(var)^+_t M(var)^-_t\right) \tag{14}$$

where MI represents the log of i index; $NC^+$ and $NC^-$ represent an increase and decrease in number of new cases; similarly, $ND^+$ and $ND^-$ are the positive and negative change in new deaths every day, respectively. $M(var)^+_t$ and $M(var)^-_t$ represent the market index variance extracted using GARCH (1,1) with increased and decreased variance in the time series.

To verify the terms of NARDL, a unit root test was performed using augmented Dickey–Fuller (ADF) and Phillips–Perron (PP) tests, and the results provided evidence of mixed order integration among the variables in Table 4. Furthermore, we also employed the Zivot–Andrews (ZA) unit root test for the structural break unit root, which gave satisfactory results. NARDL was efficient irrespective of the order of integration i.e., I(0) and/or I(1) [48].

To examine the asymmetric cointegration among the dependent and independent variables, we employed NARDL bounds tests, following [45]. We compared F-stats in the NARDL bounds test values with upper-bound and lower-bound critical values. Table 5 shows that the asymptotic F-stats fell above the upper bound critical values for both indexes, which indicates the presence of long-run cointegration among the variables [45,49]. The number of confirmed new cases and deaths was modelled separately

for each market under study, i.e., S&P Pan Arab index and S&P ESG index. The models were selected based on Akaike information criteria (AIC) with lag structures [1,0,0,1,0] for F (MI | NC+,NC−,ESGvar+, ESGvar−) and [1,0,1,0,0] for F (MI | ND+, ND−, ESGvar+, ESGvar−). This was also the case for the lag structures for the S&P conventional index, [2,9,7,9,9] and [2,1,8,7,8] for the modeling of F (MI | NC+, NC−, S&Pvar+, S&Pvar−) and F(MI | ND+, ND−, S&Pvar+, S&Pvar−) respectively. Furthermore, to treat the impact of heteroskedasticity on the third model, we adjusted for HAC (using a Newey–West estimator) until all diagnostic tests were satisfied as per the validity of the model (shown in Table 6).

**Table 5.** Unit root test of regression variables.

| | ADF | | | | PP | | | | ZA | |
|---|---|---|---|---|---|---|---|---|---|---|
| | I(0) | | I(1) | | I(0) | | I(1) | | I(0) | Break Suggested |
| | C | T & C | C | T & C | C | T & C | C | T & C | T & C | |
| Ln (new cases) | −3.1544 (0.024) | −3.4294 (0.0494) | - | - | −5.8694 (0.000) | −6.0554 (0.000) | - | - | −5.701 (0.002) | 18 July 2020 |
| Ln (new deaths) | −4.7946 (0.000) | −4.8958 (0.000) | - | - | −4.557 (0.000) | −4.0276 (0.008) | - | - | −5.308 (0.047) | 22 January 2021 |
| Ln (ESG index) | −0.7697 (0.825) | −4.3085 (0.0034) | −20.898 (0.0000) | −20.859 (0.0000) | −0.5504 (0.8778) | −6.0532 (0.0000) | −21.150 (0.000) | −21.107 (0.000) | −6.310 (0.033) | 27 May 2020 |
| Ln (S&P Index) | −0.3345 (0.9166) | −4.4948 (0.0018) | −16.189 (0.000) | −16.154 (0.0000) | −0.4540 (0.8966) | −4.9205 (0.0003) | −16.202 (0.000) | −16.154 (0.000) | −5.288 (0.021) | 15 October 2020 |
| ESG (Variance) | −7.3704 (0.000) | −7.421 (0.000) | - | - | −7.2660 (0.000) | −7.2454 (0.000) | - | - | −7.290 (0.000) | 8 August 2020 |
| S&P (Variance) | −319.34 (0.000) | −253.11 (0.000) | | | −345.73 (0.000) | −273.14 (0.000) | | | −8.481 (0.000) | 3 October 2020 |

**Table 6.** NARDL bounds test for cointegration.

| | S&P ESG Arab Index | | S&P Pan Arab Index | |
|---|---|---|---|---|
| | F (MI$_{ESG}$ | N$_{C+}$, N$_{C−}$, ESG$_{var+}$, ESG$_{var−}$) [1] | F (MI$_{ESG}$ | N$_{D+}$, N$_D$, ESG$_{var+}$, ESG$_{var−}$) [2] | F (MI$_{S\&P}$ | N$_{C+}$, N$_{C−}$, S&P$_{var+}$, S&P$_{var−}$) [3] | F (MI$_{S\&P}$ | N$_{D+}$, N$_{D−}$, S&P$_{var+}$, S&P$_{var−}$) [4] |
| | [1,0,0,1,0] | [1,0,1,0,0] | [2,9,7,9,9] | [2,1,8,7,8] |
| F-statistic | 12.46679 | 15.50373 | 5.4974 | 5.9813 |
| | I(0)   I(1) | I(0)   I(1) | I(0)   I(1) | I(0)   I(1) |
| 10% | 2.2   3.09 | 2.45   3.52 | 2.2   3.09 | 2.45   3.52 |
| 5% | 2.56   3.49 | 2.86   4.01 | 2.56   3.49 | 2.86   4.01 |
| 1% | 3.29   4.37 | 3.74   5.06 | 3.29   4.37 | 3.74   5.06 |
| | Diagnostic tests | | | |
| LM BG test | 2.4601 (0.0834) | 2.7217 (0.0643) | 1.2832 (0.2288) | 0.3035 (0.713) |
| LM BP test | 1.4275 (0.2124) | 1.4953 (0.1898) | - | 1.1716 (0.254) |
| Ramsay RESET | 1.5653 (0.2118) | 3.8654 (0.053) | 0.5772 (0.448) | 2.4254 (0.090) |

[1–4] denotes Equations (1)–(4).

Asymptotic F-stats calculated using asymmetric ARDL bounds tests were clearly greater than the upper bound critical value, which indicated that the authors should reject the null hypothesis of the bound test in favor of an alternate that ensures the presence of asymmetric cointegration, since the alternate hypothesis was accepted by following the critical values suggested by [4]. Several diagnostic tests were performed for robustness of the models along with the CUSUM test to check their stability, as shown in Figure 2. The results confirmed long-run asymmetric cointegration among the variables of interest.

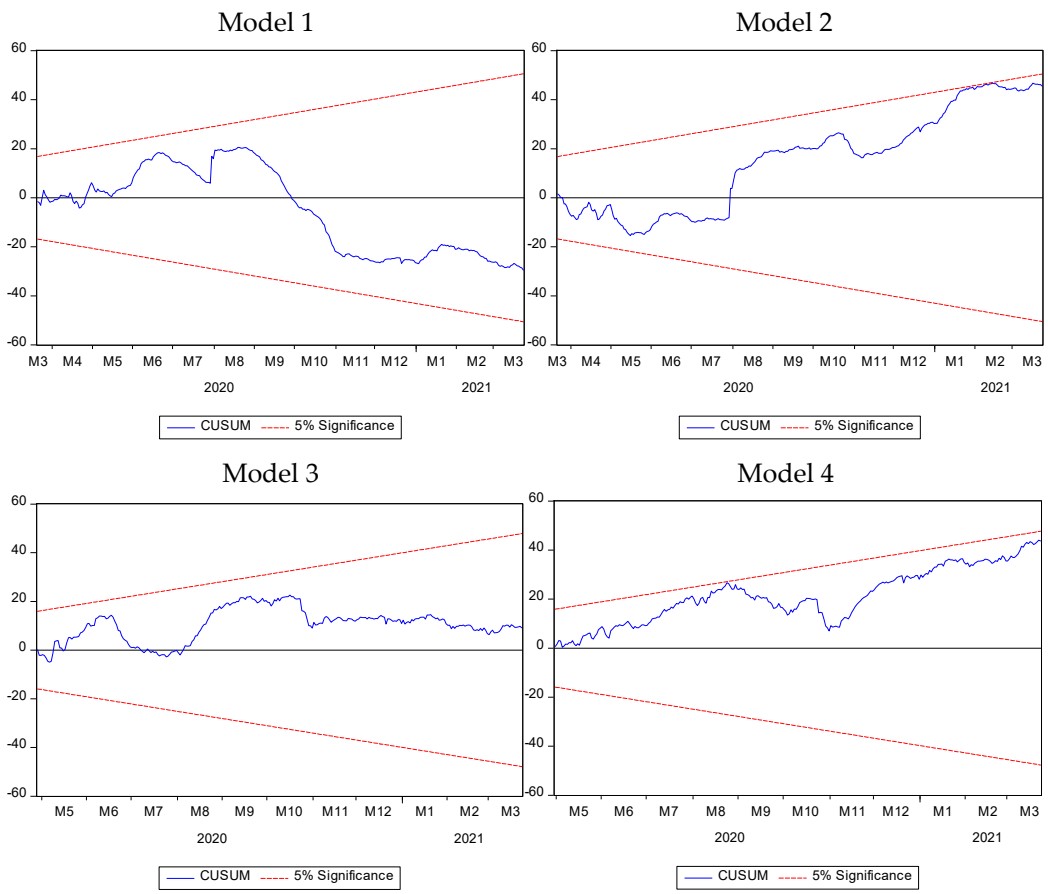

**Figure 2.** CUSUM test of stability.

Once the presence of cointegration was confirmed, we estimated the long-run asymmetric coefficients of the independent variables. Table 7 illustrates the long-run relationship between the numbers of confirmed cases and deaths and the selected indexes, with analyses of the performance of stock progress with respect to the numbers of cases and deaths. Regarding the number of new infected cases and market performance of the ESG index, increasing numbers of new cases had a positive impact on the market index, whereas an asymmetric affect was observed between increases and decreases in the number of new cases with NC+ causing the ESG market to rise with a coefficient of 0.012, i.e., significant at 10%. However, a negative relationship with far less magnitude was observed when compared to the positive relationship associated with the increase in the number of cases. Importantly, even when the number of cases decreased, the ESG index tended to rise, but the relationship was insignificant. It is worth noting that increases or decreases in the number of confirmed new cases did not affect market performance. Our results suggested that the performance of ESG did not seem to be affected that much by the number of new infected cases [32]. A similar trend was observed in our second model with respect to the number of deaths and ESG index performance. The Wald test was significant at 1%, which showed that there was strong evidence of non-linearity among the positive and negative changes in the number of deaths and the ESG market index. ND+ had a positive relationship in the long run whereas ND- was negatively associated with the ESG index. However, both long-run coefficients were not statistically significant, but rather had an inverse asymmetric effect. Our findings showed that the ESG index was not affected by the daily number of new deaths. However, in both models (1 and 2) market volatility was negatively associated with the market index. Importantly, the results gave evidence of the symmetric and linear relationship between market variance and performance. For instance, an increase in market risk would cause the market index to fall due to high volatility shock. Conversely, a decrease in market variance was found to have a positive impact on market

performance, which was primarily due to the lower risk [21]. The results suggested that the market is mostly governed by risk-averse investors, who start investing when market variance is lower than usual. By comparison, considering the S&P Pan Arab index, in the first model examining the number of confirmed new cases and the market index, the results suggested a symmetric/linear long run relationship. An increase in the number of cases (NC+) negatively affected the S&P index. The results showed that there was a negative coefficient of NC+, i.e., −0.023, significant at 10%. However, a decrease in the number of new cases (NC−) also negatively affected market performance, i.e., significant at 10%. Moreover, an increase in the number of new cases was likely to cause a decline in the market index. However, a significant rise in market performance was observed with decreases in new cases. This showed that the S&P Pan Arab index is sensitive to the number of confirmed new cases, resulting in a decline and rise in the index with an increase and decrease in new cases, respectively. This confirms the findings of prior research works regarding the negative impact of COVID-19 on market performance [17,18,26]. The second model captured the long-run impact of the number of deaths and gave similar results, but the effect of deaths was found to have an asymmetric impact on the S&P Pan Arab index. With an increase in the number of deaths (ND+) the market behaved negatively, but this was not significant. However, a decrease in number of deaths (ND−) yielded a significant rise in the stock index. For the NC− impact, the market had a long-run coefficient of −0.0192, i.e., significant at 10%. Furthermore, the Wald test, which aims to capture non-linearity, was also significant at 5%, which shows evidence of the asymmetric impact of deaths on market performance. Moreover, a symmetric impact of market volatility was observed, with negative coefficients of −191.79 and −191.92 for positive and negative changes in M(var), respectively. The coefficients were significant at 1%. The results concerning market volatility suggest that market was likely to rise if there was lower variance and vice versa. Our findings suggest that investors reacted passively as they learned of the rising number of COVID cases; this news moved the investors towards risk aversion which consequently resulted in a decline in the S&P market indexes. This indicates the role of information regarding the pandemic in supporting investment decision making. Interestingly, in our first volatility results, ESG was the less-affected market compared to the S&P index. The ESG index gradually rose irrespective of the number of cases and deaths. Although the rise of the market was not significant, it yielded a safer zone for risk-averse investors.

**Table 7.** Long-run estimates.

| | F (MI$_{ESG}$ \| N$_{C+}$, N$_{C-}$, ESG$_{var+}$, ESG$_{var-}$) [1] [1,0,0,1,0] | F (MI$_{ESG}$ \| N$_{D+}$, N$_{D}$, ESG$_{var+}$, ESG$_{var-}$) [2] [1,0,1,0,0] | F (MI$_{S\&P}$ \| N$_{C+}$, N$_{C-}$, S&P$_{var+}$, S&P$_{var-}$) [3] [2,9,7,9,9] | F (MI$_{S\&P}$ \| N$_{D+}$, N$_{D-}$, S&P$_{var+}$, S&P$_{var-}$) [4] [2,1,8,7,8] |
|---|---|---|---|---|
| NC+ | 0.0122 (1.920) [c] | - | −0.0235 (−1.943) [c] | - |
| NC− | −0.0002 (−0.0331) | - | −0.0125 (−1.758) [c] | - |
| ND+ | - | 0.010 (1.427) | - | −0.0059 (−0.615) |
| ND− | - | −0.002 (−0.243) | - | −0.0192 (−1.941) [c] |
| Mvar+ | −420.88 (−2.776) [a] | −373.28 (−2.899) [a] | −442.96 (−2.721) [a] | −191.79 (−1.830) [c] |
| Mvar− | −435.89 (−2.905) [a] | −353.05 (−2.745) [a] | −447.49 (−2.739) [a] | −191.92 (−1.830) [c] |
| Constt. | 0.424 (11.980) [a] | 0.430 (3.578) [a] | −0.124 (−0.571) | 0.191 (1.194) |
| | | Wald test for coefficient asymmetry | | |
| F-stat (Prob.) | 32.386 (0.000) | 41.68 (0.000) | 0.1203 (0.729) | 5.3391 (0.022) |
| F-stat (Prob.) | 0.6430 (0.423) | 2.161 (0.143) | 11.022 (0.001) | 0.225 (0.636) |

[1]–[4] denotes Equations (1)–(4); [a,c] denotes significance at 1%, and 10% respectively.

Table 8 represents the short-run asymmetric impact of regressors on the performance of both markets. In the short run, increases and decreases in the number of confirmed cases did not significantly affect the ESG index [50]. In addition, our results suggested a linear and insignificant relationship between NC+ and NC− and the ESG index. Furthermore, non-linear short-run relationships were found between the number of deaths and ESG performance, and the number of confirmed cases and S&P performance. Importantly, together with our long-run estimates, we observed a negative short-run effect in the

increase in market risk in the stock indexes under study [51]. Furthermore, the error correction term in all our models was negative and significant at 1%, as shown in Table 7. The negative coefficients of error correction terms show that a state of equilibrium in the long run was achieved from the state of disequilibrium in the short run.

**Table 8.** Short-run estimates.

| | F (MI$_{ESG}$ | N$_{C+}$, N$_{C-}$, ESG$_{var+}$, ESG$_{var-}$) [1] [1,0,0,1,0] | F (MI$_{ESG}$ | N$_{D+}$, N$_{D}$, ESG$_{var+}$, ESG$_{var-}$) [2] [1,0,1,0,0] | F (MI$_{S\&P}$ | N$_{C+}$, N$_{C-}$, S&P$_{var+}$, S&P$_{var-}$) [3] [2,9,7,9,9] | F (MI$_{S\&P}$ | N$_{D+}$, N$_{D-}$, S&P$_{var+}$, S&P$_{var-}$) [4] [2,1,8,7,8] |
|---|---|---|---|---|
| D(NC+) | −0.0042 (−1.317) | - | 0.0008 (0.224) | - |
| D(NC+(−1)) | - | - | 0.0075 (1.955) [c] | - |
| D(NC−) | −0.0070 (−1.296) | - | −0.008 (−1.945) [c] | - |
| D(NC−(−1)) | - | - | −0.0097 (−2.078) [b] | - |
| D(ND+) | - | 0.0074 (3.102) [a] | - | 0.0096 (3.760) [a] |
| D(ND+(−1)) | - | - | - | - |
| D(ND−) | - | −0.0031 (−0.957) | - | −0.0069 (−2.949) [a] |
| D(ND−(−1)) | - | - | - | 0.0023 (0.794) |
| D(Mvar+) | −83.291 (−5.013) [a] | −78.062 (5.779) [a] | −36.98 (−5.464) [a] | −16.188 (−5.335) [a] |
| D(Mvar+(−1)) | - | - | −4.970 (−2.532) [b] | −3.290 (−2.485) [a] |
| D(Mvar−) | 18.236 (1.388) | 19.722 (1.4832) | −37.364 (−5.428) [a] | −15.974 (−5.274) [a] |
| D(Mvar−(−1)) | - | - | −6.963 (−3.429) [a] | −4.024 (−2.833) [a] |
| CointEq(−1) | −0.0878 (−8.717) | −0.101 (−8.860) | −0.100 (−5.796) [a] | −0.0955 (−5.507) [a] |
| Wald test for coefficient asymmetry | | | | |
| F-stat (Prob.) | 0.142 (0.706) | 5.298 (0.022) | 6.0230 (0.015) | 0.2988 (0.585) |
| F-stat (Prob.) | 17.803 (0.000) | 16.841 (0.000) | 1.3362 (0.249) | 2.2749 (0.133) |

[1–4] denotes Equations (1)–(4); [a,b,c] denotes significance at 1%, 5%, and 10% respectively.

Generally, the ESG index showed lower risk after the pandemic, meaning that investors preferred this investment option and thus indicating ESG performance as an approach to alleviate risk during crises.

Our findings indicate some policy implications for governments and market regulators, since economic policies have an explicit impact on capital market performance [52]. Financial investment can play a substantial role in the reinforcement of sustainable development efforts by maintaining profit goals alongside environmental and social considerations, meaning that ESG investment could be an active tool for allocating resources in line with sustainability strategies. The findings provide sound inputs in relation to governmental policies and relief to mitigate the negative impact of the COVID-19 crisis on economic activities in general [53], and, considering the contagion in international financial markets, on the stock market in particular [54]. ESG investment supports a reduction in risks associated with uncertainty conditions during crises. Along the same lines, the better stress resilience of ESG companies in aggregate will decrease the portfolio betas or the systematic risk of the market portfolio. If enhanced ESG investment corresponds to a smaller frequency of extreme losses, then systematic risk will decrease as well. This, in turn, will reduce the alternative cost of capital for companies who wish to raise new funding from the stock markets. This might lead to a better allocation of funds within financial markets.

For investors, the findings can contribute to portfolio optimization, as including a portfolio of ESG shares can be considered a profitable investment strategy with a safe haven for overcoming market contagion.

## 5. Conclusions

This study examined the impact of the COVID-19 pandemic on the Arab capital markets by analyzing both the normal index and the ESG index. The findings indicated that COVID-19 caused a shock in both indexes. Nevertheless, the trend of the ESG index varied in the post-pandemic period with respect to the volatility level. This supports the notion that ESG investment plays a pivotal role as a safer/less risky investment channel in

times of crisis. Furthermore, the findings suggested ESG as a safe haven for investment, as it was less affected by the COVID-19 crisis compared to the normal index. Moreover, this study highlights and enriches the debate over the growing importance of ESG investment in light of critical events such as the COVID-19 crisis.

On the other hand, the findings of long-term asymmetric cointegration between the performance of the indexes under study and the number of infected cases and fatalities are an extension of studies in other markets and regions, which in turn indicates the global impact of the COVID-19 pandemic on investor behavior which was not limited to a specific region or market.

Lastly, some limitations of this study need to be mentioned, i.e., the time interval and the geographical scope of the ESG indexes. For future research, a longer time series could be studied to identify the long-term differences between normal performance and ESG performance, and other indexes or markets could be considered to support the results by applying other methodologies. Furthermore, study of sectoral differences and green companies could enhance the literature on investors' sentiment regarding risk.

**Author Contributions:** Conceptualization, M.M. and A.S.; data curation, M.M.; formal analysis, A.S.; investigation, M.M.; methodology, A.S.; resources, J.S.; supervision, J.S.; validation, A.S. and J.S.; writing—original draft, M.M. and A.S.; writing—review & editing M.M., A.S. and J.S. All authors have read and agreed to the published version of the manuscript.

**Funding:** This research received no external funding.

**Institutional Review Board Statement:** Not applicable.

**Informed Consent Statement:** Not applicable.

**Data Availability Statement:** Not applicable.

**Conflicts of Interest:** The authors declare no conflict of interest.

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
