# Peer review of "Are ESG Shares a Safe Haven during COVID-19? Evidence from the Arab Region"

_sustainability, doi:10.3390/su14010208_

Round 1

Reviewer 1 Report

Some suggestions for paper improvement:

The authors have to better justify the choice of the group of country they study. What do we learn from that countries that we do not learn from other contexts? What are the specificities of this context? Does the study take these specificities into account? How?

The authors have to compare the results of their study to those of similar empirical works in different contexts (including in the conclusions). 

The conclusions part should be extended, it has to present also:

  • the limitations of the study.
  • comparison with previous empirical findings
  • authors should provide more details on the practical/policy implications of their findings.

Reviewer 2 Report

Dear Author(s),

Please find attached the Review Report.

Reviewer 3 Report

Quality of paper is average in its current state and there is scope for improvement.  Do brainstorming, read more and conceptualise a more clear idea.

Paper needs substantial improvement. Study must have scientific approach and some data analysis.

Justify Methodology and usage of research approach. Please verify all calculations and eliminate errors if any.

Improve literature review by referring articles published in Sustainability. I ask you to refer to improve your paper:  https://doi.org/10.1108/IJOEM-04-2021-0567 ,  https://doi.org/10.1109/ICCIKE51210.2021.9410680

Structure and Flow of paper should be: Introduction, Structured Literature Review, Research Questions / Hypothesis, Research Methodology, Analysis of Data and Discussion, Findings and Conclusion.   Research questions / hypothesis must arise after Literature review.

Implement changes and resubmit paper.

Best Wishes

Round 2

Reviewer 2 Report

The amended form of the manuscript entitled “Are ESG shares a safe haven during Covid-19? Evidence from Arab region” (Manuscript ID: sustainability-1486014) enhanced its quality in a suitable manner. The author(s) considered the whole suggestions and recommendations as formulated throughout the previous review round.

Reviewer 3 Report

Authors have implemented all necessary changes.